# Multimodal LLM-guided Query Optimization for Visual-Language Retrieval

## Abstract

Vision-language retrieval (VLR), involving the use of text (or images) as queries to retrieve corresponding images (or text), has been widely used in multimedia and computer vision tasks. However, ambiguous or complex concepts contained in queries often confuse retrievers, making it difficult to effectively align these concepts with visual content, thereby limiting their performance. Existing query optimization methods neglect the feedback of retrievers' preferences, thus resulting in sub-optimal performance. Inspired by the powerful ability of Multimodal Large Language Models (MLLMs), we propose a Multimodal LLM-Guided Query Rewriter (MGQRe) for query optimization. Specifically, MGQRe first utilizes MLLM to explore the retriever's weakness and perform targeted iterative optimizations to capture the retriever's expressive preferences. Subsequently, we develop a trainable rewriter that learns this preference knowledge through a three-step tuning strategy: supervised fine-tuning, preference learning, and reinforcement learning. This ensures that the queries generated by the rewriter align with the retriever's preferences, thereby enhancing the retriever's performance. Extensive VLR benchmark experiments have demonstrated the superiority of MGQRe, as well as its generalizability and transferability. This work showcases the potential of using advanced language models to overcome the inherent limitations in current VLR technology.

## 1 Introduction

Vision-language retrieval (VLR), which involves the using of text/image as queries to retrieve corresponding image/text, has garnered significant attention from both academia and industry. Existing methods (Radford et al., 2021; Li et al., 2022; Yu et al., 2022) mainly focus on how to align text and image modalities within a shared semantic space.

Despite the progress in VLR, existing methods still face challenges on complex or ambiguous concepts in queries, due to the heterogeneity of data. For example, as illustrated in Fig 1, the retriever failed to perceive the visual features associated with the term "anticipate", thus resulting in irrelevant images. This confusion often leads to the retriever's inability to accurately align these concepts with corresponding visual features, which typically limits the performance of multimodal retrieval. To alleviate such issues, it is expected to utilize a rewriter to optimise complex concepts in the query. However, traditional rewriting methods struggle to effectively adapt queries based on the retriever's preferences, leading to suboptimal retrieval results (see Fig 1). Ensuring that the rewriter generates queries that align with the retriever's understanding preferences poses a significant challenge. Typically, exploring the retriever's preferences requires extensive human analysis and repetitive iterations to adapt queries, which is both time-consuming and costly. Multimodal large language models (MLLMs) (Wang et al., 2024; Jin et al., 2024) have demonstrated impressive analytical capabilities for addressing complex issues. Therefore, employing MLLMs as agents to capture the retriever's fine-grained preferences offers an efficient and practical solution.

Based on these observations, we propose the MLLMs-Guided Query Rewriter (MGQRe) for VLR. First, MLLMs capture the fine-grained preferences of the retriever, and a rewriter is then developed to learn preference knowledge. Specifically, we employ MLLMs as agents that mine the retriever's preferences. Based on the feedback from the retriever on the query, MLLMs continuously explore the retriever's weaknesses and iteratively optimize the query based on these weaknesses, obtaining

Figure 1: Comparison of different retrieval paradigms. (a) Direct retrieval and (b) traditional rewriting methods yield poor results because the retriever cannot accurately interpret the visual concept "anticipate". In contrast, our (c) MLLMs-Guided Query Rewriter adapts "anticipate" into a visual description that the retriever understands better, ensuring accurate retrieval.

high-quality queries that match the retriever's preferences. To distill the preference knowledge into the rewriter, we design a three-step tuning strategy: starting with Supervised Fine-Tuning (SFT) for initial warm-up, followed by Preference Rank Optimization (PRO)(Schulman et al., 2017) to align with the retriever's fine-granted preferences, and concluding with Proximal Policy Optimization (PPO)(Song et al., 2024) to further enhance the collaboration between the rewriter and the retriever. Through the above training strategies, MGQRe can generate high-quality queries that match the retriever's preferences. In summary, our contributions are as follows:

- We introduce a novel query optimizer for VLR: MLLMs-Guided Query Rewriter (MGQRe), which refines user-input queries to better align with the preferences of the retriever, thereby enhancing the alignment between the queries and the visual content and improving retrieval performance.

- We develop an automated system for constructing high-quality query datasets for VLR tasks using MLLMs. We deploy MLLMs as agents that analyze the feedback from the retriever to explore its weaknesses and iteratively optimize queries, ensuring that the refined queries align with the retriever's preferences.

- We develop a three-step learning strategy for the rewriter, consisting of Supervised Fine-Tuning (SFT), Preference Rank Optimization (PRO), and Proximal Policy Optimization (PPO). This strategy enables the rewriter to precisely adjust queries to fit the retriever's expressive preferences.

- Extensive experiments show that our method significantly outperforms other query optimization methods. Additionally, our method is generalizable and transferable, performing well across various VLR tasks.

## 2 RELATED WORK

### 2.1 VISION-LANGUAGE RETRIEVAL

In the task of VLR, the primary objective is to establish alignment between the visual and textual modalities. Previous vision-language models can be categorized into three classes: single-stream, double-stream, and dual-encoder models. Most of the **single-stream** models (Chen et al., 2020; Li et al., 2020; Kim et al., 2021) perform multi-modal interaction via self-attention alignment. These models first concatenate different modalities to produce an integrated sequence, and then perform fine-grained interaction for multi-modal alignment using the transformer's self-attention. **Double-stream** models (Li et al., 2021; 2022; Yang et al., 2022; Zeng et al., 2022) often apply the intra-modality processing along with a shared fusion encoder. This approach decouples the intra-modal and cross-modal modeling processes. They perform multi-modal interaction via the transformer's co-attention alignment, where the query vectors are from one modality, and the key and value vec-

tors are from the other. Due to the high demand for inference efficiency in visual language retrieval tasks, some scholars have proposed using the **dual-encoder** architectures for multi-modal alignment through contrastive learning (Radford et al., 2021; Xie et al., 2022; Ma et al., 2022b). In this approach, the visual embedding and text embedding are projected into the same semantic space to calculate similarity scores. Due to their efficient retrieval, dual-stream architectures are gaining more and more attention in VLR.

## 2.2 PROMPT ENGINEERING FOR VISION-LANGUAGE MODEL

Prompt engineering has become a vital technique with the rise of large pretrained models, optimizing queries into formats comprehensible by multimodal models like CLIP. Research mainly focuses on enhancing model understanding by incorporating additional knowledge, such as entity concepts from WordNet (Shen et al., 2022; Yao et al., 2022) or domain-specific insights (Ma et al., 2022a). With the emergence of LLMs, studies increasingly leverage their knowledge bases for query augmentation, including visual descriptions (Menon & Vondrick, 2022; Pratt et al., 2023) to support text-image alignment. Some research(Xie et al., 2023) also emphasizes retrieval-enhanced techniques that retrieve relevant images for cross-modal understanding. Existing approaches primarily address image classification and object detection, these coarse-grained methods struggle with complex text queries containing multiple entities and fine-grained interactions. This paper proposes a fine-grained query optimization scheme for multimodal retrieval, aimed at improving model comprehension of complex queries.

## 3 METHODOLOGY

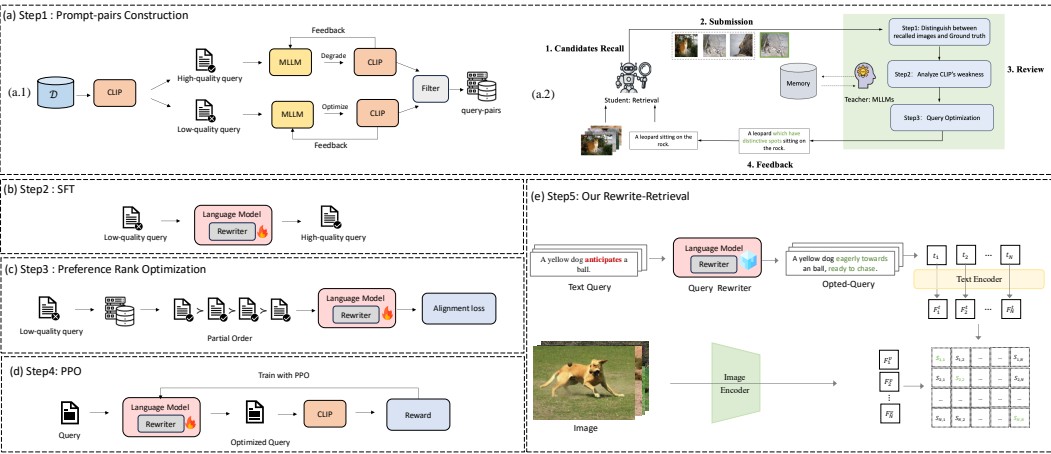

Figure 2: The framework of our method. First, we construct the high-/low-quality query pairs by the interaction of MLLMs and CLIP. Second, we conduct supervised fine-tuning (SFT) for rewriter on the query pairs. Third, we align fine-grained preferences through preference rank optimization. Fourth, we apply Proximal Policy Optimization (PPO) to the rewriter using designed rewards for further enhancement. Finally, we integrate the rewriter with CLIP to perform VLR.

Given an input query, our prompt optimizer automatically rewrites it to better match the retriever's understanding preferences, while preserving the original intent. Fig 2 provides an overview of our method, with the rewriter built upon a language model. First, we employ MLLMs to capture the retriever's preferences and gather high-quality query examples (section 3.1). These examples are then used to perform Supervised Fine-Tuning (SFT) to prepare the rewriter (section 3.2). To refine our understanding of the retriever's fine-grained preferences, we implement Preference Rank Optimization (section 3.3), followed by reinforcement learning to overcome the limitations of synthetic data (section 3.4). The trained rewriter is then integrated into the multimodal retrieval framework to perform query optimization (section 3.5).

## 3.1 DATASET CONSTRUCTION

This section outlines the collection process of data used for training the rewriter. As shown in Fig 2 (a.1), queries that align with the retriever's preferences are defined as high-quality queries. The goal of this step is to collect high-quality/low-quality query pairs with similar semantics. The data originates from Flickr30k and MSCOCO, which contain only unpaired raw queries. Initially, we categorize the queries into low and high quality based on the evaluation from the retriever.

As depicted in Fig 2 (a.2), for low-quality queries, we employ MLLMs as agents to generate high-quality queries that are easier for the retriever to understand. The specific process includes four steps: 1) Candidate Recall: retrieve candidate images using low-quality queries; 2) Submission: Submit incorrect candidates and ground truth for comparison; 3) Review: MLLMs optimize and rewrite the erroneous concepts through chain-of-thought, analyzes the differences between recalled images and correct images, identifies concepts that retriever failed to understand, and optimizes these deficiencies; 4) Feedback: Retest the rewritten queries to determine if they meet high-quality standards. This process iterates multiple turns until we get multiple high-quality queries for each low-quality query. In the above approach, MLLMs identify the retriever's shortcomings more deeply by analysing hard negative samples of the retriever, thus more accurately capturing and understanding the retrieval model's expression preferences.

Similarly, for high-quality queries, we blur concepts using MLLMs and simplify the rewriting of queries with MLLMs to decrease their similarity to images, thereby generating corresponding low-quality queries. Finally, we calculate the textual similarity (Reimers, 2019) between low and high-quality queries and filter out examples with low similarity. Detailed implementation specifics and prompt templates can be found in the Appendix A.1.

## 3.2 SUPERVISED FINE-TUNING

With the low-quality/high-quality query pairs established, we can train a query optimizer to develop basic query optimization capabilities. A parallel query corpus consists of low-quality query $x$ and high-quality query $y$, referred to as the $D_{SFT}$. If a low-quality query $x$ corresponds to multiple high-quality queries, we take the highest-quality query as $y$. $\pi(\cdot)$ and $\theta$ denote the query rewriter to be trained during SFT and its parameters, which can be any pretrained language model. We train the rewriter by minimizing the cross-entropy loss. The training objective for SFT is to optimize the following loss function:

$$L_{\text{SFT}}(\theta) = -\mathbb{E}_{(x,y) \sim D_{SFT}} \sum_t \log \pi \left( y_t | y_{<t}, x; \theta \right) \tag{1}$$

SFT can be viewed as a warm-up phase, and thus, the effectiveness of the supervised fine-tuning model is generally moderate. To further enhance model performance, we proceed with preference optimization.

## 3.3 PREFERENCE OPTIMIZATION

To enhance the rewriter's understanding of the model's fine-grained conceptual preferences, we performed preference optimization. This process requires constructing a dedicated preference dataset $D_{PRO}$. As described in Appendix A.1, we generate multiple high-quality queries for each low-quality query and obtain text-image similarity scores from the retrieval system, which serve as rewards for preference learning. These scores allow us to rank the enhanced queries from high to low. To minimize bias from the reward model and enhance fine-grained preference comparisons from a global perspective, we introduce Preference Rank Optimization (PRO) based on the Bradley-Terry model(Song et al., 2024). This method guides the model to learn the ranking of rewrites according to feedback from the retriever. According to the Bradley-Terry model, the probability of choosing a policy is proportional to its corresponding reward. Given the partial order relation $y_1 \succ y_2$, the preference probability can be expressed as:

$$P_{BT} = \frac{\exp(r(y_1, x))}{\exp(r(y_1, x)) + \exp(r(y_2, x))} \tag{2}$$

where $r(\cdot)$ is the reward function, which is defined as the normalized log probability of the rewrite generated in PRO. PRO extends pairwise partial order into general listwise partial order. The PRO

loss is expressed by the equation:

$$L_{\text{PRO}}(\theta) = -\mathbb{E}_{(\mathbf{x},\mathbf{y}) \sim D_{PRO}} \sum_{j=1}^{k-1} \log \frac{\exp\left(\frac{\pi_{\text{PRO}}(y_j|x;\theta)}{T_j^j}\right)}{\sum_{i=j}^{k} \exp\left(\frac{\pi_{\text{PRO}}(y_i|x;\theta)}{T_j^i}\right)} \tag{3}$$

where $T_j^i = \frac{1}{r(y_j)-r(y_i)}$ and $T_j^j = \min_{i>j}(T_j^i)$ are used to measure ranking difference. $k$ denotes the number of candidate high-quality queries, $\pi_{\text{PRO}}$ and $\theta$ refer to the policy model and its parameters.

### 3.4 REINFORCEMENT LEARNING

Due to the limited datasets collected and inherent noise, relying solely on constructing query pairs is insufficient for effectively guiding the rewriter. Consequently, we propose using reinforcement learning to enable the rewriter to explore freely and better adapt to the retriever. Initially, we define the rewards for reinforcement learning, focusing on the improvement in cross-modal similarity as a measure of query quality. The reward is then defined as follows:

$$r(x,y) = 20 * (s_{clip}(y,v_x) - s_{clip}(x,v_x)) \tag{4}$$

where $s_{clip}(\cdot)$ denotes the similarly score between query and image, $v_x$ represents the ground image related to query $x$. After obtaining the predefined reward, we suggest using Proximal Policy Optimization (PPO) during reinforcement learning training to enhance our retriever. The PPO algorithm directly optimizes the expected reward:

$$L_{\text{PPO}}(\theta) = -\mathbb{E}_{\mathbf{x} \sim D_{PPO}, \mathbf{y} \sim \pi_{\text{PPO}}(\cdot|x)}[r(x,y) - \beta \cdot \log \frac{\pi_{\text{PPO}}(y|x)}{\pi_{\text{PRO}}(y|x)}] \tag{5}$$

Following (Ziegler et al., 2019), we adopt an adaptive KL penalty strategy with parameter $\beta$, which is used to prevent the policy from deviating too far from the initial distribution $\pi_{\text{PRO}}$.

### 3.5 INTEGRATION OF THE REWRITER FOR VLR

In our work, we select the simple yet effective dual-encoder model CLIP as the foundational model (a detailed introduction and training for CLIP can be found in Appendix A.2). We incorporate the trained rewriter into the CLIP framework, as illustrated in Fig 2 (e). During CLIP fine-tuning, we freeze the rewriter and randomly perform query rewriting with probability $p$. For inference, given an input text query, we utilize the rewriter to generate the opted-query. This opted-query is modeled through the text encoder to obtain text feature, which is then computed with the features of candidate images for similarity score. Finally, images are recalled in descending order of similarity.

## 4 EXPERIMENTS

### 4.1 EXPERIMENT SETTING

#### 4.1.1 DATASETS

We primarily evaluate on two benchmark datasets: Flickr30k and MSCOCO, to validate its effectiveness. (1) **Flickr30k** (Plummer et al., 2015) contains 31,000 images, each with 5 captions, and is divided into 29K/1K/1K images for training, validation, and testing, respectively (Li et al., 2021). (2) **MSCOCO** (Lin et al., 2014) consists of 123,287 images, also with 5 captions each, and is split into 114K/5K/5K for training, validation, and testing. To further assess the transferability of our method, we evaluate its performance on other visual-text retrieval datasets, including (3) **MSR-VTT** (Xu et al., 2016), which includes 10,000 videos with a total of 200,000 text descriptions. We utilize 9K videos for training and evaluation on a 1K test set. Additionally, (4) **SBU30k** (Ordonez et al., 2011) contains 36,000 image-text pairs randomly sampled from SBU Captions, divided into 30K/3K/3K for training, validation, and testing. Furthermore, we randomly sample from CC12M (Changpinyo et al., 2021) and YFCC15M (Thomee et al., 2016) to obtain (5) **CC30K** and (6) **YFCC30K**. Details on the query pair dataset constructed for the rewriter can be seen in Appendix A.1.

### 4.1.2 BASELINES

We will validate our method on advanced dual-encoder retrieval models, specifically: (1) **CLIP**(Radford et al., 2021), a powerful dual-encoder model pre-trained through contrastive learning; (2) **CoCa**(Yu et al., 2022), a framework that integrates various pre-training paradigms, utilizing its image encoder and unimodal text decoder for retrieval; and (3) **EVA-02-CLIP** (Sun et al., 2023), which employs novel representation learning techniques to enhance CLIP's performance.

Our approach will be compared with current query optimization methods, including: (1) **Det-CLIP** (Yao et al., 2022): An entity knowledge enhancement method that integrates WordNet conceptual knowledge into query entities. (2) **CLIP-GPT** (Maniparambil et al., 2023) An entity description enhancement scheme that incorporates visual descriptions generated by LLMs into query entities. (3) **RACLIP** (Xie et al., 2023): A retrieval augmentation approach that uses relevant images to enrich the query with cross-modal semantics. (4) **LLMsRewrite**: A description rewriting scheme that utilizes LLMs for query optimization. We have designed templates to guide LLMs, which include task descriptions and crafted examples.

To evaluate the effectiveness of our dataset generation strategy, we compared three dataset construction scenarios: (1) **Direct Generation Strategy**: Queries are generated directly based on image content using MLLMs. (2) **Feedback Enhancement Strategy**: MLLMs generate queries based on the image, then refine these queries using the in-retriever similarity score between the generated query and the image as a metric for feedback and subsequent optimization. (3) **Challenge Response Strategy**: Our implementation involves using challenging negative and positive samples from the retriever to identify its weaknesses, allowing for continuous query adjustment and optimization.

In addition, we explore various open-source LLMs for rewriters, including Qwen-7B (Bai et al., 2023) ("Qwen-7B-chat"), Baichuan2-7B (Yang et al., 2023) ("Baichuan2-7B-chat"), Llama (Touvron et al., 2023) ("Llama2-7B-chat"), and Vicuna (Chiang et al., 2023) ("vicuna-7B-v1.3").

### 4.1.3 IMPLEMENTATION DETAILS

We fine-tune a pre-trained retrieval (e.g., CLIP) directly without the pretraining, making the process lightweight. The settings for fine-tuning the retrieval are as follows: we use the Adam optimizer with a weight decay of 1e-3 and a batch size of 256. The total number of fine-tuning epochs is set to 20. The initial learning rate is 1e-6, with a cosine learning rate scheduler, and a warm-up strategy is applied for the first 2k steps. The probability of random rewriting during retrieval fine-tuning $p$ is set to 0.6. In our experiments, the visual encoder includes two variants: Vision Transformer (ViT-B/32, ViT-B/16, and ViT-L/14) and ResNet (RN50 and RN101). For the text encoder, we use the vanilla Transformer from CLIP (Vaswani et al., 2017). Input images are resized to $224 \times 224$, and input sequences are truncated or padded to 77 tokens.

For the prompt-pairs construction, we use GPT-4V as the MLLM. The details for constructing low-quality/high-quality query pairs can be found in Appendix A.1. Unless otherwise specified, we use Llama2-7b as the query rewriter. For the SFT phase, we set the learning rate to 1e-5, the batch size to 32, and run for 10 epochs. In the DPO phase, the learning rate is set to 5e-7, with a batch size of 16, across 5 epochs, and a rank length of 5. For the PPO phase, CLIP with ViT-B/32 is used for reward calculation. The learning rate is set to 5e-6, with a batch size of 32, and 1 epoch of fine-tuning. The KL coefficient $\beta$ is set to 0.1. Following previous work (Radford et al., 2021), we use recall $R@h(h = 1, 5, 10)$ as the evaluation metric.

### 4.2 MAIN RESULT

We conduct evaluations of our method on two benchmark datasets, Flickr30K and MSCOCO, utilizing advanced VLR dual-encoder frameworks. As shown in Table 1, we compare various query optimization methods, analyzing the results to draw several conclusions:

**Entity Enhancement Surpass Image Retrieval Enhancements**: Our experiments demonstrate that various query optimization methods enhance vision-language retrieval performance. Notably, methods incorporating entity visual descriptions (CLIP-GPT) and entity knowledge (DetCLIP) significantly outperform enhancements based on related images (RACLIP). We speculate that because the description of entities and knowledge are more granular than the global information provided by

Table 1: Fine-tuning results for image-text retrieval on the Flickr30K (1K) test set and MSCOCO (5K) test set. Notations: V-Encoder: vision encoder; # PT Data: the pre-training datasets.

| Methods | V-Encoder | # PT Data | Flickr30K(1K) | | | | | | MSCOCO(5K) | | | | | |
| | | | I2T Retrieval | | | T2I Retrieval | | | I2T Retrieval | | | T2I Retrieval | | |
| | | | R@1 | R@5 | R@10 | R@1 | R@5 | R@10 | R@1 | R@5 | R@10 | R@1 | R@5 | R@10 |
| CLIP | ViT-B/32 | NA | 64.8 | 85.7 | 92.5 | 49.2 | 79.3 | 86.8 | 43.7 | 73.5 | 82.6 | 32.7 | 63.3 | 75.0 |
| DetCLIP | ViT-B/32 | NA | 65.2 | 86.3 | 93.5 | 50.7 | 79.2 | 86.8 | 45.2 | 73.7 | 83.4 | 33.4 | 63.5 | 75.0 |
| CLIP-GPT | ViT-B/32 | NA | 66.5 | 88.1 | 93.6 | 51.2 | 80.1 | **87.8** | 46.1 | 74.0 | 83.7 | 34.1 | 63.7 | 75.3 |
| RACLIP | ViT-B/32 | NA | 65.1 | 86.2 | 93.0 | 50.2 | 79.4 | 87.1 | 45.1 | 73.6 | 82.7 | 33.1 | 63.4 | 74.9 |
| LLMsRewrite | ViT-B/32 | NA | 66.5 | 87.6 | 93.3 | 50.8 | 79.8 | 87.3 | 45.7 | 73.9 | 83.5 | 33.5 | 63.5 | 75.1 |
| MGQRe | ViT-B/32 | NA | **67.1** | **88.7** | **94.2** | **52.2** | **80.6** | 87.7 | **46.7** | **74.6** | **84.3** | **35.2** | **64.4** | **75.8** |
| CLIP | ViT-B/32 | Laion400M | 89.1 | 97.8 | 98.9 | 74.1 | 92.6 | 95.9 | 65.3 | 85.9 | 91.9 | 48.1 | 75.0 | 83.7 |
| DetCLIP | ViT-B/32 | Laion400M | 89.2 | 97.8 | 99.1 | 74.6 | 92.8 | 96.0 | 65.5 | 85.9 | 92.1 | 48.3 | 75.1 | 83.7 |
| CLIP-GPT | ViT-B/32 | Laion400M | 89.7 | 98.7 | 99.2 | 75.2 | 93.1 | 96.1 | 66.2 | 86.2 | 92.3 | 48.8 | 75.3 | 84.3 |
| RACLIP | ViT-B/32 | Laion400M | 89.2 | 98.0 | 98.9 | 74.4 | 92.8 | 96.0 | 65.3 | 86.1 | 92.1 | 48.5 | 75.2 | 83.8 |
| LLMsRewrite | ViT-B/32 | Laion400M | 89.3 | 98.1 | 99.0 | 74.5 | 92.9 | 96.0 | 65.3 | 86.0 | 91.9 | 48.3 | 75.2 | 84.0 |
| MGQRe | ViT-B/32 | Laion400M | **90.8** | **99.2** | **99.6** | **75.7** | **93.6** | **96.5** | **66.8** | **86.9** | **92.7** | **49.5** | **76.0** | **84.6** |
| CoCa | ViT-B/32 | Laion-2B | 85.5 | 96.5 | 98.7 | 72.0 | 91.2 | 95.4 | 63.9 | 85.6 | 91.0 | 45.6 | 72.1 | 82.2 |
| DetCoCa | ViT-B/32 | Laion-2B | 85.6 | 96.5 | 98.7 | 72.2 | 91.2 | 95.4 | 63.8 | 85.5 | 91.0 | 45.8 | 72.1 | 82.1 |
| CoCa-GPT | ViT-B/32 | Laion-2B | 86.2 | 97.0 | 98.8 | 72.2 | **91.6** | 95.3 | 64.3 | 85.7 | 91.0 | 46.0 | 72.2 | 82.3 |
| RACoCa | ViT-B/32 | Laion-2B | 85.8 | 96.6 | 98.8 | 72.1 | 91.2 | 95.3 | 64.1 | 85.6 | 91.1 | 45.7 | 72.1 | 82.0 |
| LLMsRewrite | ViT-B/32 | Laion-2B | 86.1 | 96.7 | 98.8 | 72.1 | 91.3 | 95.5 | 64.2 | 85.6 | 91.1 | 45.8 | 72.1 | 82.2 |
| MGQRe | ViT-B/32 | Laion-2B | **86.8** | **97.3** | **98.9** | **72.7** | 91.6 | **95.8** | **65.0** | 85.9 | 91.5 | 46.6 | **72.7** | 82.5 |
| EVA-02-CLIP | ViT-B/16 | Merged-2B | 90.8 | 98.7 | 99.2 | 78.9 | 94.7 | 97.0 | 69.1 | 89.2 | 94.0 | 52.6 | 78.5 | 86.8 |
| DetEVA-02-CLIP | ViT-B/16 | Merged-2B | 90.9 | 98.6 | 99.1 | 79.1 | 94.6 | 97.0 | 69.3 | 89.2 | 94.0 | 52.7 | 78.5 | 86.7 |
| EVA-02-CLIP-GPT | ViT-B/16 | Merged-2B | 91.1 | 98.7 | 99.2 | 79.3 | 94.7 | 97.1 | 69.4 | 89.3 | 94.3 | 52.6 | 78.6 | 86.8 |
| RAEVA-02-CLIP | ViT-B/16 | Merged-2B | 90.7 | 98.6 | 99.1 | 79.0 | 94.6 | 97.0 | 69.1 | 89.0 | 94.0 | 52.6 | 78.5 | 86.6 |
| LLMsRewrite | ViT-B/16 | Merged-2B | 91.0 | 98.6 | 99.2 | 79.2 | 94.7 | 97.0 | 69.2 | 89.2 | 94.1 | 52.8 | 78.6 | 86.8 |
| MGQRe | ViT-B/16 | Merged-2B | **91.5** | **98.7** | **99.5** | **79.7** | **95.0** | **97.3** | **69.9** | **89.8** | **94.4** | **53.6** | **79.1** | **87.2** |

images, they better capture the fine-grained cues required for text-image alignment. In VLR, visual description of entities proves more advantageous than conceptual knowledge, because it provides more detailed perceptual information that helps the model capture specific visual details.

**MGQRe Outperforms All Existing Approaches**: As shown in Table 1, our method (MGQRe) improves the retrieve's performance best. Compared to entity enhancement methods that provide only entity knowledge and retrieval enhancement methods that provide global coarse-grained perception, our approach can optimize fine-grained concepts within queries, covering not only entities but also interactive and descriptive concepts. Therefore, our method displays superior performance in multimodal retrieval tasks.

Unoptimized rewriters (LLMsRewrite) show underperformance in VLR. The main reason is that these rewriters do not adjust based on retriever's feedback, thus generating queries that may not align with the retriever's understand preferences, and may even distort the original queries' intent. MGQRe, learning from retriever's preferences, identifies which expressions are more effective for retrievers and performs targeted optimizations, thus enhancing queries' quality.

**Significant Improvements Across Different Retrievers**: Further experiments on CLIP-like models, detailed in Table 1, demonstrated that models like CoCa and EVA-02-CLIP achieved significant performance improvements on most metrics after adopting our method. This underscores our approach's generalizability and effectiveness. Furthermore, in Appendix A.3, we show that our method is applicable to retrievers with various vision encoders.

## 4.3 ABLATION STUDY

Using CLIP with ViT-B/32, pre-trained on Laion400M, as our baseline, we conducted comprehensive ablation studies on MGQRe to evaluate the impact of data collection, training strategies, and large language models on performance.

Table 2: Ablation studies on dataset collection strategies. The Fine-tuning dataset is Flickr30k.

| Methods | I2T Retrieval | | T2I Retrieval | |
| | R@1 | R@5 | R@1 | R@5 |
| Baseline | 89.1 | 97.8 | 74.1 | 92.6 |
| Direct Generation | 89.2 | 98.1 | 74.5 | 92.8 |
| Feedback Enhancement | 90.5 | 98.8 | 75.1 | 93.4 |
| Challenge Response | **90.8** | **99.2** | **75.7** | **93.6** |

Table 3: Ablation studies on training strategies. The Fine-tuning dataset is Flickr30k.

| Methods | I2T Retrieval | | T2I Retrieval | |
| | R@1 | R@5 | R@1 | R@5 |
| Freeze | 89.3 | 98.1 | 74.5 | 92.9 |
| SFT | 89.6 | 98.4 | 75.0 | 93.3 |
| SFT + PRO | 90.4 | 98.9 | 75.3 | **93.6** |
| SFT + PRO + PPO | **90.8** | **99.2** | **75.7** | 93.6 |

### 4.3.1 Data Collection Strategies

In this section, we evaluate different data generation strategies. As shown in Table 2, Direct Generation ignores the preferences of the retrieval model, resulting in limited performance improvements due to the lack of task-specific tuning. Feedback Enhancement considers the retrieval model's preferences during query rewriting but fails to explore the model's weaknesses, limiting its effectiveness in more challenging scenarios. Challenge Response leverages MLLMs to analyze the model's weaknesses by focusing on difficult negative and positive samples, enabling targeted optimization to address these shortcomings and further enhance overall performance.

### 4.3.2 Training Strategies

We conducted ablation studies on different training strategies to assess their impact on the performance of our rewriter. The experimental results are shown in Table 3.

Initially, using only Supervised Fine-Tuning (SFT), the model demonstrated basic rewriting capabilities and adapted to preliminary query optimization. However, this method had its limitations as it relied heavily on existing data, preventing the model from fully understanding the retriever's fine-grained preferences and thus limiting performance improvements. By incorporating Preference Rank Optimization (PRO), the model gained a deeper understanding of the retriever's detailed preferences, enabling the generation of higher-quality queries and significantly enhanced retrieval performance. Finally, the introduction of reinforcement learning overcame the limitations of data, allowing the model to dynamically explore and adapt to the retriever's preferences, further improving performance. This progression illustrates the effectiveness of layering advanced learning strategies to significantly boost the capabilities of the query rewriter.

Table 4: Ablation studies on language models. The Fine-tuning dataset is Flickr30k.

| Methods | LLMs | I2T Retrieval | | | T2I Retrieval | | |
|---|---|---|---|---|---|---|---|
| | | R@1 | R@5 | R@10 | R@1 | R@5 | R@10 |
| CLIP | NA | 89.1 | 97.8 | 98.9 | 74.1 | 92.6 | 95.9 |
| MGQRe | Vicuna-7B | 90.5 | 99.1 | 99.3 | 75.2 | 93.3 | 96.4 |
| | Baichuan2-7B | 90.5 | 99.0 | 99.5 | 75.5 | 93.3 | **96.5** |
| | Qwen-7B | 90.6 | 98.9 | 99.4 | 75.6 | 93.4 | 96.4 |
| | Llama2-7B | 90.8 | 99.2 | **99.6** | 75.7 | 93.6 | 96.5 |
| | Llama2-13B | **90.9** | **99.2** | 99.6 | **75.8** | **93.7** | 96.5 |

Table 5: Performance on various VLR datasets.

| Data | Methods | I2T Retrieval | | | T2I/T2V Retrieval | | |
|---|---|---|---|---|---|---|---|
| | | R@1 | R@5 | R@10 | R@1 | R@5 | R@10 |
| MSR-VTT | CLIP | - | - | - | 34.6 | 63.1 | 73.7 |
| | MGQRe | - | - | - | **35.8** | **63.8** | **74.7** |
| CC30k | CLIP | 59.0 | 81.2 | 88.3 | 58.5 | 80.5 | 87.1 |
| | MGQRe | **61.2** | **81.8** | **88.5** | **59.8** | **81.5** | **87.6** |
| SBU30k | CLIP | 43.8 | 66.3 | 74.3 | 43.4 | 65.7 | 74.4 |
| | MGQRe | **45.2** | **67.7** | **75.3** | **44.7** | **67.3** | **74.6** |
| YFCC30k | CLIP | 37.4 | 56.3 | 64.3 | 35.8 | 56.6 | **64.6** |
| | MGQRe | **39.1** | **56.7** | **64.7** | **36.6** | **57.2** | 64.4 |

### 4.3.3 Large Language Models

We conduct ablation studies using various large language models as rewriters to assess the capabilities and limitations of query editing techniques. As shown in Table 4, the performance across all tested LLMs is relatively consistent, with LLama2 showing significantly better results. Importantly, the performance of LLama2-7B is comparable to that of LLama2-13B, suggesting that under our training strategy, smaller language models are sufficiently effective for query optimization tasks without significantly impacting efficiency. This finding highlights the potential for optimising retrieval efficiency without compromising the quality of queries.

### 4.4 Transferable of Rewriter

We further explore the transferability of our rewriter across various visual language retrieval datasets. As shown in Table 5, we perform experimental validations on the video-text retrieval dataset MSR-VTT as well as other image-text retrieval datasets. Despite the rewriter being trained on the Flickr30K and MSCOCO datasets, it still enhances the performance of the retrievers on these different datasets.

This result shows that the model preferences and optimisation capabilities learned by the rewriter from a specific dataset are transferable. These capabilities can be effectively transferred to other visual language retrieval tasks, indicating that the rewriter has broad applicability in improving the retrieval performance of diverse visual language datasets.

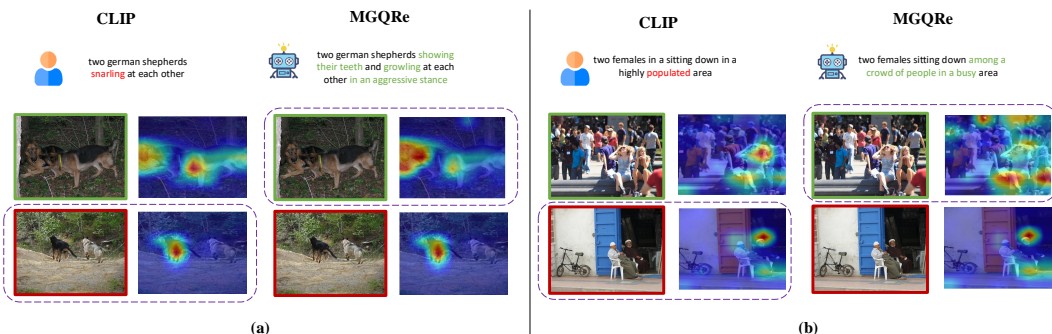

Figure 3: Visualization example of text-to-image retrieval, showcasing heatmaps corresponding to different text queries. The image inside the dashed box is the retrieved result of the query. MGQRe optimizes user-input queries into queries that are more comprehensible to the retriever, facilitating better alignment between the queries and the visual content of images.

## 4.5 VISUALIZATION ANALYSIS

To gain a deeper understanding of how MGQRe improves the matching of queries to images, we have visually demonstrated that MGQRe helps the retriever to focus on image regions relevant to the query semantics. We utilize the Integrated Gradients algorithm (Qi et al., 2019), which calculates and evaluates the impact of each feature on the final prediction.

As shown in Fig 3 (a), the heatmap clearly indicates that CLIP struggles with the visual concept of "snarling", as it fails to match "snarling" with the corresponding visual content in the image, leading to inaccurate retrieval results. MGQRe optimizes the query by rewriting "snarling" into a more retriever-friendly visual description, such as "showing their teeth". This helps the retriever better understand the concept and focus its attention on the relevant visual content, resulting in accurate retrieval. Similarly, Fig 3 (b) demonstrates that CLIP also has difficulty aligning the term "populated" with its corresponding visual content. MGQRe optimizes the query into a description that fits CLIP's visual preferences, aiding the retriever in more accurately aligning the query with the image content. This visualization underscores MGQRe's effectiveness in improving retrieval accuracy by refining queries to match the retriever's comprehension preferences.

## 4.6 LIMITATIONS

While MGQRe effectively rewrites queries to align better with the preferences of the retriever, it can occasionally introduce additional information that may act as noise in multimodal matching, potentially affecting the performance of cross-modal alignment. Addressing this issue is a key direction for future optimization. Additionally, MGQRe faces challenges in retrieval efficiency. We believe that with the rapid advancement in large language models, the emergence of more lightweight, high-performance language models (Hu et al., 2024) will gradually mitigate this issue.

## 5 CONCLUSION

In this paper, we introduce a query optimizer named MGQRe for visual-text retrieval, designed to rewrite query concepts that are difficult for retrievers to understand into expressions that align with their comprehension preferences. Specifically, we utilize multimodal large language models to capture the preferences of retrievers, analyze their performance weaknesses, and iteratively optimize to generate high-quality queries that meet these preferences. We then implement a three-phase optimization strategy that effectively distills the retriever's preferences into the rewriter, ensuring that user-input queries are optimized into forms more easily understood by the retriever. Extensive experimental results demonstrate that our method outperforms other query optimization strategies, showcasing strong generalizability and transferability. These contributions provide valuable insights for future research in visual-text retrieval. Future research directions could further explore how to integrate user preferences and domain-specific knowledge into the rewriter to enrich the application scenarios of query optimization in multimodal retrieval.

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
