# A APPENDIX

## A.1 THE LOW-QUALITY/HIGH-QUALITY QUERY PAIRS CONSTRUCTION

As illustrated in Fig 2 (a.1), our corpus $D$ comprises approximately 150k queries and their corresponding 30k images extracted from the Flickr30k and COCO training sets, totaling around 300k queries and 100k images. During the data construction process, we chose CLIP with ViT-B/32 as the base retrieval. Initially, we assessed the quality of the queries based on their performance in CLIP retrieval: queries whose corresponding Ground Truth (GT) is not within the top 10 recall results are considered low-quality, indicating they exceed the retriever's comprehension capability; queries with GT in the top three results are deemed high-quality. Ultimately, we identified about 150k low-quality queries and 20k high-quality queries.

For low-quality queries, we employ an iterative optimization method as shown in Fig 2 (a.2). The retriever inputs the top 3 retrieved images into the multimodal large model for analysis and optimization. When the optimized query's similarity exceeds that of the original low-quality query, it is added to the corresponding high-quality query collection. For each low-quality query, we aim to obtain at least 5 high-quality queries. Finally, for each low-quality query, we store the corresponding high-quality queries and their similarity scores for subsequent Preference Rank Optimization (PRO).

For high-quality queries, we generate low-quality queries by utilizing MLLMs to perform concept obfuscation. When a generated query's similarity score falls below that of the original high-quality query, it is designated as the corresponding low-quality query. For each high-quality query, we generate one low-quality query. In the subsequent filtering stage, we not only filter out mismatched low/high-quality query pairs but also exclude samples that cannot generate corresponding query pairs after 10 rounds of iteration, as well as those filtered out by rule-based anomaly detection.

Ultimately, we obtain 150,000 low/high-quality query pairs, with 130,000 as original low-quality queries, each associated with multiple high-quality queries and corresponding scores, and 20,000 as original high-quality queries. The prompt templates used in the MLLMs are shown in Fig 4, with specific interaction examples available for reference.

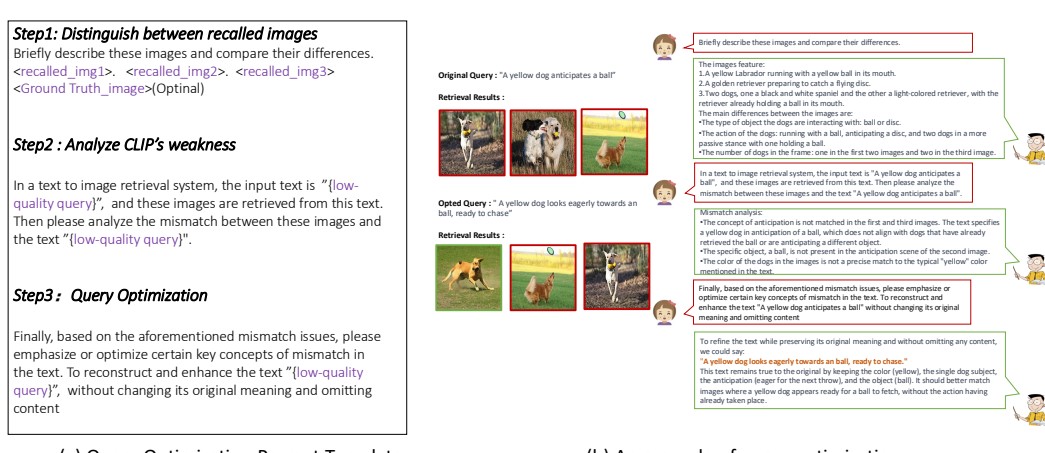

(a) Query Optimization Prompt Template      (b) An example of query optimization

Figure 4: Instruction template and interactive example for the multimodal large language model GPT-4v used to optimise low-quality queries.

## A.2 DUAL-ENCODER FRAMEWORK CLIP

As shown in Figure 2 (e), images and texts are encoded by an image encoder and a text encoder respectively, then projected into the same semantic space for effective retrieval. Formally, assuming we have $N$ samples in a batch, $B = \{(v_i, t_i)\}_{i=1}^{N}$ denotes the training dataset, where $(v_i, t_i)$ is the $i$-th image-text pair. The matched image-text pairs are considered positive samples, while other

pairwise combinations serve as negative samples. We define the image-to-text contrastive loss as:

$$
\begin{aligned}
L_{i2t} &= -\frac{1}{N} \sum_{(v_i, t_i) \in B} y \cdot \log p(v_i, t_i) \\
&= -\frac{1}{N} \sum_{(v_i, t_i) \in B} \log \frac{\exp\left(F_i^v \cdot F_i^t / \tau\right)}{\sum_{j=0}^{N} \exp\left(F_i^v \cdot F_j^t / \tau\right)}.
\end{aligned}
\tag{6}
$$

where $F_i^v$ and $F_i^t$ are the normalized embedding of $v_i$ and $t_i$. $\tau$ is the temperature hyper-parameter. Similarly, we can define the text-to-image contrastive loss as:

$$
L_{t2i} = -\frac{1}{N} \sum_{(v_i, t_i) \in B} \log \frac{\exp\left(F_i^v \cdot F_i^t / \tau\right)}{\sum_{j=0}^{N} \exp\left(F_j^v \cdot F_i^t / \tau\right)}.
\tag{7}
$$

The final contrastive loss can be denoted as:

$$
L = L_{i2t} + L_{t2i}
\tag{8}
$$

The dual-encoder framework aligns images and text using global features, but it lacks fine-grained cues for precise vision-language alignment.

## A.3 ABLATION STUDIED ON VISION ENCODER

We fine-tune various vision encoders on Flickr30K, and the experimental results are shown in Table 6. MGQRe boosts the performance on both ResNet (He et al., 2016) (RN50, RN101) and ViT (Han et al., 2022) (ViT-B/16, ViT-L/14) vision backbones. Despite the powerful ViT-L/14 encoder having mastered rich multi-modal semantic information, MGQRe can still improve its performance. This suggests that our approach is an encoder-independent boosting method.

Table 6: Fine-tuning results on various CLIP vision encoders. The Fine-tuning dataset is Flickr30k.

| V-Encoder | # PT Data | Methods | I2T Retrieval R@1 | I2T Retrieval R@5 | T2I Retrieval R@1 | T2I Retrieval R@5 |
|---|---|---|---|---|---|---|
| RN50 | CC12M | CLIP | 73.4 | 91.3 | 55.0 | 81.7 |
| | | MGQRe | **75.6** | **92.8** | **56.5** | **82.0** |
| RN101 | YFCC15M | CLIP | 76.1 | 93.2 | 56.8 | 85.0 |
| | | MGQRe | **78.8** | **94.0** | **59.5** | **86.2** |
| ViT-B/16 | Laion400M | CLIP | 89.8 | 97.8 | 75.8 | 93.2 |
| | | MGQRe | **91.1** | **98.5** | **76.2** | **93.7** |
| ViT-L/14 | Laion400M | CLIP | 92.3 | 99.3 | 79.9 | **95.3** |
| | | MGQRe | **92.7** | **99.5** | **80.5** | 95.2 |