# OpenReview forum: "Multimodal LLM-guided Query Optimization for Visual-Language Retrieval"
_ICLR.cc/2025/Conference — ICLR 2025 Conference Withdrawn Submission_

### Official Review · Reviewer_d6cF · 2024-10-30

**Soundness:** 2
**Presentation:** 3
**Contribution:** 2
**Rating:** 5
**Confidence:** 4

**Summary:**

This paper presents a method called MGQRe (Multimodal LLM-Guided Query Rewriter), which leverages multimodal large language models (MLLMs) to perform targeted iterative optimizations. Previous approaches in vision-language retrieval utilize query rewriters to refine complex concepts within the query, but they often fall short in generating queries that include more direct and relevant information. MGQRe addresses this gap by iteratively incorporating the retriever's preference knowledge using MLLMs. To achieve this, MGQRe employs supervised fine-tuning, preference rank optimization, and proximal policy optimization. As a result, MGQRe significantly outperforms previous query optimization methods.

**Strengths:**

This paper (MGQRe) shows three major contributions (strengths):

1. First MLLM-based query optimization: MGQRe introduces to refine user queries to better align with retriever preferences.

2. Automated dataset construction: MLLM is utilized to construct high-quality query datasets, where MLLM acts as agents, iteratively refining queries based on retriever feedback to address its weaknesses and preferences.

3. Effective training strategy & state-of-the-art results:  Supervised fine-tuning, preference rank optimization, and proximal policy optimization —enables MGQRe to precisely tailor queries to meet retriever requirements and achieves SOTA.

**Weaknesses:**

1.Lack of novelty: Query rewriting is not a particularly novel approach and has been widely used within the retrieval community. In particular, LLMs are already prevalently used to enhance text queries. Furthermore, some of the proposed training strategies for the rewriter do not appear highly effective. As shown in Table 3, adding PRO and PPO improves performance by only around 1% for I2T retrieval and 0.7% for T2I retrieval.

2.Limited performance gain: While MGQRe does achieve a decent performance improvement compared to CLIP and its variants, the average performance gain over other LLM-based query rewriting methods is marginal. This outcome raises concerns about efficiency, especially considering that MGQRe leverages MLLMs, which are significantly more complex than standard LLMs.

**Questions:**

The proposed method is evaluated on MS COCO and Flickr, which are relatively small-scale datasets where recent models already achieve near-perfect recall scores. Are there any more complex evaluation setups available to better validate MGQRe's superiority?

---

### Official Review · Reviewer_RUBt · 2024-11-01

**Soundness:** 2
**Presentation:** 2
**Contribution:** 2
**Rating:** 3
**Confidence:** 4

**Summary:**

This paper introduces a query optimizer named MGQRe for visual-text retrieval, designed to rewrite query concepts that are difficult for retrievers to understand into expressions that align with their comprehension preferences. Experimental results are conducted on several benchmarks to show the effectiveness of the proposed method.

**Strengths:**

+ A query optimizer called MGQRe is proposed for visual-text retrieval.
+ A three-step learning strategy is devised for the proposed rewriter.
+ The effectiveness of the proposed rewriter is partially evaluated.

**Weaknesses:**

- The technical novelty of the proposed rewriter is very limited. SFT, PRO, and PPO are directly borrowed from previous work. I'm not excited by the algorithm modifications.
- According to Table 1, the improvements caused by the proposed rewriter are generally not significant. I tend to think that the proposed rewriter is not that effective.
- Since the retriever's preferences are used in PRO by the proposed rewriter, it seems that the comparison in Table 1 is not fair.

**Questions:**

Please see my questions in Weaknesses.

---

### Official Review · Reviewer_n4Yo · 2024-11-03

**Soundness:** 3
**Presentation:** 2
**Contribution:** 2
**Rating:** 6
**Confidence:** 2

**Summary:**

This paper introduces **MGQRe**, a method for aligning Large Language Models (LLMs) in query rewriting to enhance visual language retrieval. The model is trained through a three-stage process:

- **Supervised Fine-Tuning (SFT)**: MGQRe uses high- and low-quality query pairs for each image, generated iteratively by an MLLM. This process refines or degrades query quality under guided adjustments, creating a dataset of paired queries for training.

- **Preference Rank Optimization (PRO)**: Multiple queries for each image are ranked based on feedback from visual retrieval models, allowing MGQRe to learn nuanced ranking preferences and align more closely with the retrieval model’s decision patterns.

- **Proximal Policy Optimization (PPO) with Reinforcement Learning**: The visual retrieval model acts as a reward model, guiding MGQRe through reinforcement learning to adaptively improve query quality based on retrieval performance.

**Evaluation and Results**: Tested across various visual retrieval benchmarks, MGQRe shows improvement in retrieval accuracy, validating the effectiveness of its query rewriting approach within visual-language retrieval tasks.

**Strengths:**

+ The focus on improving query rewriting through LLMs has practical application value. If consistently effective, this approach could be beneficial across a range of existing visual retrieval systems.

+ Using CLIP or other dual-stream visual retrieval models as reward functions for query rewriting is a sound approach, facilitating straightforward environment setup for PPO and ranking generation for PRO.

+ The paper provides detailed steps for training data collection, which enhances reproducibility and could support implementation efforts by others in the field.

+ The method demonstrates improved retrieval accuracy across multiple benchmarks, indicating its potential for broad application.

**Weaknesses:**

- **Lack of Alternatives for Training Stages**: No alternative methods are tested for any of the three training stages (SFT, PRO, PPO), limiting insight into whether this specific combination is uniquely effective or if other configurations could achieve similar results.

- **Marginal Improvement in Accuracy**: The observed increase in search accuracy across benchmarks is relatively modest. The method appears to have less impact on performance compared to changes in the rewrite LLM or the visual encoding model. An analysis of implementation and inference costs could help justify the value of these incremental gains.

- **Absence of Open-Source Code**: Given the three-stage process with distinct learning objectives, the lack of open-sourced code may pose challenges to reproducing results, potentially limiting broader adoption and evaluation by the community.

**Questions:**

My questions given the above weakness are:

1. how can we validate the proposed combination to be uniquely effective?

2. how to justify the relatively modest improvement tin search accuracy?

---

### Official Review · Reviewer_bFVF · 2024-11-03

**Soundness:** 3
**Presentation:** 1
**Contribution:** 2
**Rating:** 5
**Confidence:** 3

**Summary:**

The paper addresses Vision-Language Retrieval (VLR) by optimizing user queries and improving their quality, to increase retrieval results. The authors introduce the Multimodal LLM-Guided Query Rewriter (MGQRe), which aims to fit the retrieval model (e.g, CLIP) preferences. To this end, the authors present a three-step learning approach for the rewriter (SFT, PRO and PPO) allowing MGQRe to align refined queries closely with the retriever’s preferences.

**Strengths:**

- The paper introduces a new approach for optimizing VLR queries by aligning them with the retriever’s model preferences.
- Extensive experiments are conducted with multiple backbone models and datasets, demonstrating the method’s robustness, including applications to Video-Text Retrieval tasks.
- The authors present a thorough ablation study on several key components of the method, providing insights into the impact of each element on the overall performance.

**Weaknesses:**

Motivation:
- In lines 37-40, Fig 1:  The example about “failing to perceive the visual features associated with the term ‘anticipate’” lacks detail. Which image corpus was used for retrieval? What were the top-5 results, and which model was used? Without these specifics, it’s hard to confirm the issue described here (I have to take the authors’ word for it). It’s worth noting that even a standard CLIP model performs well on the COCO validation set, with ground truth in the top-1 for around 43% of queries (Table 1). Such failures may reflect a fine-grained performance limitation rather than a fundamental flaw.
- Building on the previous point, the paper would benefit from additional examples with their top-k (e.g., k=5,10) retrieval results. It is not fully clear to me what the method aims to achieve: is it enhancing the clarity of the user’s query? rephrasing it to better align with CLIP’s language/understanding? maybe adding more details to the user's query? More examples would help clarify what specifically drives the observed improvements, which is related to my next points about the evaluation.

Evaluation:

- I have a major concern regarding the data the method was trained on. The data constructed on section 3.3 is based on two datasets (COCO and Flickr30k), which is used for the PRO stage. Therefore, the method is not comparable to others presented in Table 1. Authors should compare apple-to-apples, by evaluating different methods that were trained on the same supervised benchmarks. For example, if the process already involve the training sets of COCO and Flickr30k, maybe a CLIP model that was trained on both (an alternative to the proposed method) will perform best? this point is essential to understand if the observed improvements come from the method itself.
- In addition, the authors use different CLIP models in different phases (described in Sec 4.1.3), which is type of model assemble. If the method uses the feature vectors of both CLIP-ViT-B/32 and CLIP-ViT-B/16 models, maybe the vanilla baseline of concatenating their features will perform best? While I am not suggesting the authors test this baseline, it would strengthen the method’s consistency to use a single model version across all parts.

Presentation:

The writing and presentation require substantial improvement and clarification to meet ICLR standards. As a reader, there were numerous points where the authors’ intentions were unclear to me. Some claims lack citations, several equations need more explanation, and there are minor issues throughout. Below are specific areas for the authors to address:

- Lines 45-56: “Typically, …  requires extensive human analysis and repetitive iterations to adapt queries” - please add citations.
- Section 2.2: There are multiple missing citations. Claims like “has become a vital technique” or “Existing approaches primarily address…”  for example, need supporting citations to provide evidence. Without references, the reader is left relying on the authors’ word.
- Figure 2: The text is too small to read, even when enlarged by 300%. Please fix the font size for readability.
- Line 169: Up to this point, it is unclear what is meant by a “low/high quality”query. Line 166 suggests it may involve “similar semantics”, but this needs clarification.
- Eq. 1 needs more clarification. What is y_t? (Line 189 states that if there are multiple y, you choose only 1). What does it mean a log of a high-quality query (log \pi)? It is not clear.
- Line 204: The authors should briefly explain the process for generating low/high quality queries here, rather than only referring to the appendix. This would improve the continuity of the text.
- There is no appendix attached to the original paper. The authors submitted it to the supplementary material field instead.
- Eq 5 has a typo (I assume), otherwise the log yield zero. What is the right term?
- Line 297: The authors mean they fine-tuned pre-trained CLIP? Please rephrase.
- Section 4.6 (Limitations):  It is helpful that the authors discuss limitations, but these should be expanded upon, maybe with a reference to the Appendix. For example, the statement “GQRe faces challenges in retrieval efficiency” could be elaborated, as understanding the method’s overhead is important for readers evaluating whether the performance gain justifies the cost. Please consider this point.

**Questions:**

- Lines 50-78: Could you clarify what is meant by “retriever preferences”? Isn’t this simply the model’s behavior? Also, how does this relate to user preferences?
- Line 167: What do you mean by "unpaired" queries? The next sentence states that you categorize them based on the retriever evaluation (which uses pairs). I am confused.

---

### Note · Authors · 2024-12-03

I have read and agree with the venue's withdrawal policy on behalf of myself and my co-authors.